# Chondrogenic Potential of Human Adipose-Derived Mesenchymal Stromal Cells in Steam Sterilized Gelatin/Chitosan/Polyvinyl Alcohol Hydrogels

**DOI:** 10.3390/polym15193938

**Published:** 2023-09-29

**Authors:** Mario Alberto Pérez-Díaz, Erick Jesús Martínez-Colin, Maykel González-Torres, Carmina Ortega-Sánchez, Roberto Sánchez-Sánchez, Josselin Delgado-Meza, Fernando Machado-Bistraín, Valentín Martínez-López, David Giraldo, Érik Agustín Márquez-Gutiérrez, Jorge Armando Jiménez-Ávalos, Zaira Yunuen García-Carvajal, Yaaziel Melgarejo-Ramírez

**Affiliations:** 1Laboratorio de Biotecnología, Unidad de Gerociencias, Instituto Nacional de Rehabilitación Luis Guillermo Ibarra Ibarra, Ciudad de México 14389, Mexico; marioprz0586@gmail.com (M.A.P.-D.); maykel.gonzalez@conahcyt.mx (M.G.-T.); minaorsa@hotmail.com (C.O.-S.); joss.endorf05@gmail.com (J.D.-M.); machado_bistrain@hotmail.com (F.M.-B.); 2Departamento de Farmacobiología, Centro de Investigación y de Estudios Avanzados, Ciudad de México 07360, Mexico; 3Unidad de Ingeniería de Tejidos, Terapia Celular y Medicina Regenerativa, Instituto Nacional de Rehabilitación LGII, Ciudad de México 14389, Mexico; sanchez2.roberto@gmail.com (R.S.-S.);; 4Department of Cell and Tissue Biology, School of Medicine, Universidad Nacional Autónoma de México, Ciudad de México 04510, Mexico; davidgiraldo@comunidad.unam.mx; 5Cirugía Plástica y Reconstructiva, Centro Nacional de Investigación y Atención de Quemados, Instituto Nacional de Rehabilitación LGII, Ciudad de México 14389, Mexico; dr_erikmarquez@hotmail.com; 6Unidad de Biotecnología Médica y Farmacéutica, Centro de Investigación y Asistencia en Tecnología y Diseño del Estado de Jalisco, A.C. (CIATEJ), Guadalajara 44270, Mexico; avalos.joar@gmail.com

**Keywords:** gelatin, chitosan, polyvinyl alcohol, chondrogenic potential, hydrogels, human adipose-derived mesenchymal stromal cells, biocompatibility

## Abstract

Cross-linked polymer blends from natural compounds, namely gelatin (Gel), chitosan (CS), and synthetic poly (vinyl alcohol) (PVA), have received increasing scrutiny because of their versatility, biocompatibility, and ease of use for tissue engineering. Previously, Gel/CS/PVA [1:1:1] hydrogel produced via the freeze-drying process presented enhanced mechanical properties. This study aimed to investigate the biocompatibility and chondrogenic potential of a steam-sterilized Gel/CS/PVA hydrogel using differentiation of human adipose-derived mesenchymal stromal cells (AD-hMSC) and cartilage marker expression. AD-hMSC displayed fibroblast-like morphology, 90% viability, and 69% proliferative potential. Mesenchymal profiles CD73 (98.3%), CD90 (98.6%), CD105 (97.0%), CD34 (1.11%), CD45 (0.27%), HLA-DR (0.24%); as well as multilineage potential, were confirmed. Chondrogenic differentiation of AD-hMSC in monolayer revealed the formation of cartilaginous nodules composed of glycosaminoglycans after 21 days. Compared to nonstimulated cells, hMSC-derived chondrocytes shifted the expression of CD49a from 2.82% to 40.6%, CD49e from 51.4% to 92.2%, CD54 from 9.66 to 37.2%, and CD151 from 45.1% to 75.8%. When cultured onto Gel/CS/PVA hydrogel during chondrogenic stimulation, AD-hMSC changed to polygonal morphology, and chondrogenic nodules increased by day 15, six days earlier than monolayer-differentiated cells. SEM analysis showed that hMSC-derived chondrocytes adhered to the surface with extended filopodia and abundant ECM formation. Chondrogenic nodules were positive for aggrecan and type II collagen, two of the most abundant components in cartilage. This study supports the biocompatibility of AD-hMSC onto steam-sterilized GE/CS/PVA hydrogels and its improved potential for chondrocyte differentiation. Hydrogel properties were not altered after steam sterilization, which is relevant for biosafety and biomedical purposes.

## 1. Introduction

Cross-linked polymer blends from natural compounds, namely gelatin (Gel) and chitosan (CS), have received increasing scrutiny in the last decade because of the versatility of their structure, biocompatibility, and ease of use for tissue engineering and pharmaceutical formulation [1]. These polymer networks are known to be chemical or permanent hydrogels with various physical shapes. Previous studies have shown the successful synthesis and characterization of genipin CS/Gel interpenetrating polymer network (IPN) hydrogel [2]. The alginate-CS hydrogel, integrated with gel microspheres containing tetracycline hydrochloride and prepared using the emulsion cross-linking method, showed effective antibacterial properties [3]. Biomimetic photo-induced cross-linked layered Gel-CS hydrogel showed remarkable recovery in an in vivo osteochondral defect study [4]. The blend of Gel-CS and a controlled-rate freezing technique resulted in three-dimensional (3D) scaffolds that supported mouse embryonic fibroblasts efficiently [5]. The biomimetic bilayer structure of the pads is a promising feature for tissue engineering [6]. Using CS suspension mixed with Gel hydrogel allows the preparation of membranes that support human MG-63 osteoblast adhesion [7]. The literature also reported that using glutaraldehyde as a crosslinker agent elicited the fabrication and control of the pore size of a new Gel-CS composite hydrogel with proven fibroblast viability and proliferation properties [8]. There again, CS and Gel hydrogel served to obtain a 3D scaffold sandwiched with a self-assembled polycaprolactone (PCL), claimed for use as a cardiac patch [9].

Furthermore, the polyelectrolyte Gel-CS hydrogel strategy was employed to formulate an optimized platform for 3D bioprinting and implement the resulting constructs in skin tissue engineering for wound healing [10]. Moreover, hydrogel sheets of CS, Gel, and honey were produced as burn wound dressings [11]. Another approach addressing the need for new wound-healing and regenerative materials is the radiation-induced crosslinking of Gel and carboxymethyl chitosan (CM-CS) to produce hybrid hydrogels [12]. On the other hand, the feasibility and appropriateness of developing new materials from CS and Gel along with other compounds have also been explored to address the shortcomings of the use of Gel hydrogel derivatives, including degradation rate problems, low flexibility, and brittleness [13,14]. Polyvinyl alcohol (PVA) is an example that represents one such initiative because of its nontoxicity and biocompatibility. It has been combined with CS and Gel to produce a hydrogel Gel/CS/PVA by gamma radiation-induced polymerization for wound dressing use [15]. Earlier works also refer to the use of PVA associated with CS, polyacrylic acid (PAAc), polyhydroxy propyl methacrylate (PHPMA), and Gel to yield a pH-responsive hydrogel [16], hydrogels of PVA grafted with succinic acid [17], and a blended hydrogel by esterification of PVA with Gel for usage as drug delivery systems [18]. Recently, the freeze-drying process was applied to synthesize Gel/CS/PVA hydrogels with different polymer ratios and their subsequent assessment of their rheological, thermal stability, and cytotoxic features [19]. It was found that Gel/CS/PVA presented enhanced mechanical properties and was a good candidate for tissue engineering. However, to the best of our knowledge, the cell therapy applications of this hydrogel have not been explored to any extent. This study aims to investigate the biocompatibility and chondrogenic potential of employing the differentiation of human adipose-derived mesenchymal stromal cells (AD-hMSC) cultured on Gel/CS/PVA hydrogels. The hMSC-derived chondrocytes cultured onto hydrogels can be used as covering biomaterials to promote the regeneration of cartilage and other tissues or as hMSC carriers in injured areas. Collectively, the results of this research demonstrate that Gel/CS/PVA hydrogels are biocompatible and show significant potential for use in cartilage tissue engineering techniques.

## 2. Materials and Methods

### 2.1. Chemicals and Materials

Gelatin from bovine skin (Gel; type B, Bloom ~75), low molecular weight chitosan (CS; 92.2% degree of deacetylation), and polyvinyl alcohol (PVA; 89 kDa, 99.8% hydrolysis) were purchased from Sigma Aldrich (St. Louis, MO, USA). For multilineage differentiation: transforming growth factor-β (TGF-β; Peprotech, London, UK. Cat. 100-21), bone morphogenetic protein 4 (BMP-4; Peprotech, Cat. 120-05), dexamethasone, trypsin and ascorbic acid (Gibco, Carlsbad, CA, USA). For enzymatic digestion: type II collagenase (Worthington Biochemical, Lakewood, NJ, USA). For viability assays: live/dead viability/cytotoxicity kit for mammalian cells (Invitrogen L3224, Carlsbad, CA, USA).

### 2.2. Gel/CS/PVA Hydrogel Preparation

The hydrogel films were prepared by the solvent-casting method according to the methodology reported by Rodriguez-Rodriguez et al. (2019), with modifications [19]. A 2.5% *w*/*v* Gel solution was prepared by adding gelatin to distilled water and mechanically stirring at 37 °C for 2 h. CS was dissolved in a 3% *v*/*v* acetic acid solution at a concentration of 2.5% *w*/*v* under magnetic stirring for 24 h at 37 °C. PVA polymer was dissolved in distilled water at 90 °C with stirring for 3 h until a 2.5% *w*/*v* solution was obtained. Polymer solutions were mixed in a 1:1:1 (*w*/*w*) polymer ratio of Gel:CS:PVA, respectively, under magnetic stirring for 3 h at 20 °C to form a homogeneous mixture. The polymer mixture (8 g) was transferred into plastic Petri dishes (60 mm × 13 mm) and allowed to dry in a convection oven (Luzeren, Hangzhou, China) at 50 °C for 48 h. The formed dried, film-like hydrogels were then peeled from the Petri dishes and neutralized by immersion in a NaOH 0.1 N aqueous solution in distilled water for 30 min, and then washed thrice with deionization to remove any remaining acetic acid traces. Next, the hydrogels were irradiated using ultraviolet radiation at 254 nm at 120,000 μJ/cm^2^ for 60 min in a CL-1000 UV Crosslinker (Cambridge, UK) for the physical cross-linking process. Finally, they were sterilized in phosphate-buffered saline (PBS) using a steam autoclave SE 510 (Yamato Scientific, Tokyo, Japan) at 121 °C and 103.4 KPa for 15 min [20]. The protocol followed for the preparation of hydrogels is outlined in Figure A1.

### 2.3. Scanning Electron Microscopy (SEM)

The microarchitecture and topology of the Gel/CS/PVA hydrogel; as well as the cell adherence and morphology in chondrogenic constructs, were observed by SEM. Briefly, all samples were fixed in 3% (*v*/*v*) glutaraldehyde in a buffering solution of 0.1 M sodium cacodylate (pH 7.2) for 48 h (16538, Electron Microscopy Science, Hatfield, PA, USA). Subsequently, the samples were dehydrated in ethanol ranging from 30% to 99.99% for 30 min each (E7148, Sigma-Aldrich, Merck, Darmstadt, Germany). Then, they were dried using a critical point dryer CO_2_ chamber (K850, Quorum Technologies, Lewes, UK), as previously reported. Finally, all dried samples were mounted in 25 mm diameter aluminum pin stubs and mounted on them using 25-mm-diameter carbon-based electrically conductive carbon double-sided adhesive discs (Agar Scientific, Stansted, UK). The samples were coated with gold in two thin layers in two cycles of 20 sec each by plasma-assisted deposition using a manual sputter coater (AGB7340, Agar Scientific, Stansted, UK). The images were acquired with a FIB-SEM Crossbeam 550 field emission microscope Zeiss (Oberkochen, Baden-Württemberg, Germany) at 4 kV.

### 2.4. Atomic Force Microscopy (AFM)

The topography was determined using an atomic force microscope (AFM; 5420 microscope, Agilent Technologies, Santa Clara, CA, USA). Gel/CS/PVA hydrogel films were dehydrated at a controlled temperature of 50 °C in a convection oven (Luzeren, Hangzhou, China) and later placed in a desiccator until observation to avoid hydration due to relative humidity. For each sample, three random regions were examined with a scan size of 4 × 4 μm^2^. Measurements were performed in contact mode under ambient conditions with silicon tips (CSC11, Silicon-MDT, Moscow, Russia) taking images at a scanning rate of 3 lines/s. Moreover, at least three samples were measured for each kind of Gel/CS/PVA hydrogel. The AFM images were analyzed with WSxM 4.0 free software (Develop 8.6).

### 2.5. Isolation, Viability, and Proliferation of Human AD-hMSC

After informed consent was obtained, AD-hMSCs were isolated from adipose tissue obtained through elective reconstructive surgeries or liposuction. The tissue was transported under sterile conditions (sterile glass flask with DMEM-F12 and 10% of penicillin-streptomycin) and maintained at 4 °C until processing. Briefly, 15–20 g of tissue was mechanically dissected into pieces of 1 cm^3^ and washed with 1% of penicillin-streptomycin/phosphate-buffered saline (PBS). The tissue was digested with a 100 mg/mL type II collagenase (Worthington) solution in an orbital shaker at 200 rpm and 37 °C for 45 min. Following this, the stromal vascular fraction (SVF) was collected, inactivated, filtered, and centrifuged at 1500 rpm, and the pellet was suspended in supplemented medium (10% FBS/1% GlutaMAX/1% sodium pyruvate/1% penicillin-streptomycin in DMEM-F12). Cells were seeded at a 2.0 × 10^5^ cells/cm^2^ density in culture flasks. After 24 h, nonadherent cells were discarded by washing culture flasks and replenished with a fresh medium until 70–80% confluence was reached. Adherent cells were detached and subcultured using trypsin-EDTA 0.05% and were used for further experiments. AD-hMSC viability was determined using a live/dead viability/cytotoxicity kit for mammalian cells (Invitrogen L3224, Carlsbad, CA, USA). Monolayer cells or chondrogenic constructs were incubated in 4 mM calcein solution and 1 mM ethidium homodimer (EthD-1) for 45 min. For proliferation analysis, AD-hMSCs were immunolabeled with a primary monoclonal antibody against the human nuclear protein Ki67 (1:50, Santa Cruz Biotechnology, Dallas, TX, USA). Nuclei were counterstained with 1 mg/mL of 4′, 6-diamidine-2-phenylidole-dihydrochloride (DAPI). After incubation, cells or chondrogenic constructs were washed and observed under epifluorescence microscopy with an Axio Observer A.1 microscope and a confocal laser scanning microscope LSM800 (Carl Zeiss, Oberkochen, Baden-Würtemberg, Germany).

### 2.6. Multilineage Differentiation of AD-hMSC

The differentiation potential of AD-hMSCs was evaluated using a combination of high-density cultures and growth factor-enriched differentiation media. For chondrogenic differentiation, an initial density of 4 × 10^4^ cells/cm^2^ was seeded in culture flasks and allowed to settle in supplemented growth media (DMEM-F12, 1% FBS, 1% penicillin-streptomycin). After 24 h, the supplemented media was replaced with fresh chondrogenic induction media (10 ng/mL TGF-β1, 1.0 mg/mL insulin, 10 mM dexamethasone, 20 mg/mL ascorbic acid, 1% FBS, and 1% penicillin-streptomycin in DMEM-F12). For adipogenic differentiation, 3.0 × 10^4^ cells/cm^2^ were cultured in the supplemented medium for 24 h and replaced with induction medium (1 µM dexamethasone, 1.7 µM insulin, 1% FBS, and 1% penicillin-streptomycin in DMEM-F12). Finally, for osteogenic differentiation, 2.0 × 10^5^ cells/cm^2^ were plated and cultured for 24 h, and the medium was replaced with induction media (10 ng/mL BMP-4, 100 nM dexamethasone, 50 µM ascorbic acid, 1% FBS, and 1% penicillin-streptomycin in DMEM-F12). Differentiation assays were carried out for 21 days before the histological staining assessment.

### 2.7. Flow Cytometry for Mesenchymal and Chondrogenic Markers

The expression of surface markers and phenotype were analyzed by flow cytometry with mesenchymal and cartilage-specific antibodies. Confluent AD-hMSC or chondrogenic-differentiated cells were harvested and counted. A total of 2.5 × 10^5^ cells per tube were washed with 2 mL of PBS, centrifuged at 1500 rpm for 10 min, and resuspended in 1 mL of FACS flow solution. Then, the cells were labeled using 1.5 µg of CD marker antibodies and incubated for 30–45 min at 4 °C without light. The mesenchymal phenotype was detected with CD34 (Becton Dickinson, Franklin Lakes, NJ, USA, 550761), CD45 (BD, 555482), CD73 (BD, 560847), CD90 (BD, 555595), CD105 (BD, 560839), CD166 (BD, 559263), and HLA-DR (BD, 559868) antibodies. In contrast, chondrogenic differentiated cells were detected with CD49a (BD, 559596), CD49e (BD, 555617), CD54 (BD, 559771), CD99 (BD, 555688), CD151 (BD, 556057), and HLA-DR/DP/DQ (BD, 555558). After incubation, the cells were washed in 2 mL of PBS to remove antibody excess, centrifuged at 1500 rpm, and suspended in 1 mL of PBS. Surface marker expression was detected using a FACSAria III cell cytometer (Becton Dickinson, Franklin Lakes, NJ, USA).

### 2.8. Histological Staining

Mesenchymal stromal cell-derived chondrocytes (hMSC-derived chondrocytes) were stained with alcian blue for glycosaminoglycan detection. Cells were fixated on Khale solution for 20 min, washed with PBS, and stained with alcian blue for 12 h before being washed with 0.1 N HCl twice. Finally, the cells were gradually embedded in 50%, 75%, and absolute glycerol before visualization. Osteogenic differentiation was determined using Von Kossa staining for calcium deposits. Cells were fixed with 10% formalin for 10 min. After that, formalin was washed twice with distilled water. Then, 5% silver nitrate (AgNO_3_) solution was added and exposed to UV light for 1 h, washed, and replaced with sodium thiosulfate solution. Oil red staining was used for visualization of adipogenic differentiation. Cells were fixed with 10% formalin for 10 min and then washed twice with distilled water. Preheated oil red solution (1:1) was added for 10 min and washed 3 times with distilled water. Positive staining was visualized under conventional microscopy in an AxioObserver A.1 microscope (Carl Zeiss).

### 2.9. Chondrogenic Gel/CS/PVA Constructs and Analysis

To determine the biocompatibility and chondrogenic potential, AD-hMSC were cultured on Gel/CS/PVA hydrogels, referred to as chondrogenic constructs. Cells were expanded and directly seeded onto 15-mm-diameter steam-sterilized Gel/CS/PVA hydrogels at a density of 4 × 10^4^ cells/cm^2^. After 24 h, the basic supplemented media was replaced with fresh chondrogenic induction media. In situ, chondrogenic stimulation continued for 21 days, and the medium was changed periodically. The control group was determined to be AD-hMSC nonstimulated with chondrogenic induction medium and instead cultured in basic supplemented DMEM-F12/1% FBS medium. Cell viability, adherence, and morphology of hMSC-derived chondrocytes in chondrogenic constructs were analyzed as previously described.

### 2.10. Immunochemistry Assay for Cartilage Markers

The expression of cartilage markers in chondrogenic constructs was determined using immunofluorescence analysis against human aggrecan (ACAN) and type II collagen (COL2). Constructs were fixed in 2% paraformaldehyde for 20 min, then washed with PBS and blocked (PBS, 1% FBS, 0.1% Triton X-100) for 2 h. Incubation with primary antibodies against ACAN (1:50) and COL2 (1:50) was carried out overnight at 4 °C. After incubation, the antibody solution was discarded, and the constructs were washed and incubated in a secondary antibody solution (1:250) for 2 h. Finally, constructs were counterstained with 1 mg/mL of 4′, 6-diamidine-2-phenylidole-dihydrochloride (DAPI) to stain nuclei and visualized under epifluorescence microscopy with an AxioObserver A.1 microscope (Carl Zeiss).

### 2.11. Statistical Analysis

Experiments were repeated at least three times. Individual cell counts determined cell viability and mortality percentages in five fields per experiment. Data are expressed as mean ± standard deviation (SD). *p*-values below 0.05 were considered significant. Cell cytometry for surface marker expression analysis was performed with CellQuest Pro v 5.2.1 software (BD San Jose, CA, USA). Briefly, 10,000 events per sample were acquired, analyzed, and graphed with the software FlowJo 10.9.0 (BD San Jose, CA, USA). Histograms for each marker showed overlays of populations and percentages of negative control and positive cells: control in gray, APC in red, PE in orange, and FITC in green. Statistical analysis was performed using Graph-Pad Prism 6.0.

## 3. Results and Discussion

For this study, UV-crosslinked Gel/CS/PVA hydrogels were produced and steam sterilized. Biocompatibility with AD-hMSCs and the chondrogenic potential of hMSC-derived chondrocytes were analyzed.

### 3.1. Synthesis of Gel/CS/PV Hydrogel, Surface Topology, and Morphology Analysis

In this study, we aimed to prepare hydrogel films from Gel/CS/PVA mixture neutralized with NaOH 1 N to act as scaffolds to investigate the biocompatibility and chondrogenic potential through the differentiation of AD-hMSC. SEM microphotographs of Gel/CS/PVA hydrogel samples showed thicknesses between 20 µm and 81 µm (Figure 1a–c). Two-and three-dimensional (2D-3D) AFM images of Gel/CS/PVA hydrogel in the dry state. The 2D AFM images showed the brightest areas, indicating the highest point of the membrane surface, and the dark regions show valleys. The 3D reconstruction surface predominantly showed a microscale topography structured into hills and valleys. Randomly distributed aggregates of different sizes are observed throughout the hydrogel film, increasing the surface roughness and producing an average roughness (Ra) of 33.2 nm. The cross-sectional surface roughness profile confirms its roughness and uneven surface due to undissolved polymer particles during hydrogel preparation (Figure 1d).

Gelatin is a derivative of collagen used for the synthesis of hydrogels that retain adhesion peptides and other cues to promote cell interaction [21]. Gelatin can be remodeled by mesenchymal stromal cells to modify the ECM environment and promote cartilage regeneration via chondrogenesis [22]. It has been reported that chitosan and gelatin hydrogels have no cytotoxic activity against stem cells or somatic cells; however, it was necessary to synthesize a Gel/CS/PVA hydrogel resembling the physical characteristics of noninjured cartilage extracellular matrix (ECM) [23]. Our results show the importance of generating a support biomaterial under controlled conditions with topological features to improve cell adhesion and interaction with stem cells to activate chondrogenesis. Further studies will be required to understand how this feature regulates hMSC differentiation.

### 3.2. Human Adipose-Derived Stromal Cells Fulfill Mesenchymal Criteria

AD-hMSC isolated from SVF displayed fibroblast-like morphology through in vitro cell expansion. These cells formed a monolayer on the culture dish surface that reached approximately 70% confluence by day 15 of culture (Figure 2a–c), while calcein analysis showed 96% cell viability and 4% mortality under standard conditions (Figure 2d). Replicative capacity was evaluated via the expression of the nuclear protein Ki67. It was possible to detect the Ki67 signal in the nuclei of AD-hMSC at different time points, but in a regular pattern (Figure 2e,f) colocalized with DAPI (Figure 2g) in 69% of the total cells (Figure 2h). Notably, cells retained spiculated morphology during in vitro expansion. Once the viability and proliferation of adherent cells were determined, the mesenchymal phenotype was analyzed through surface marker expression. Flow cytometry analysis revealed expression for CD73 (98.3%), CD90 (98.6%), and CD105 (97.0%). The hematopoietic markers were CD34 (1.11%), CD45 (0.27%), and HLA-DR (0.24%) (Figure 2i). To verify the multilineage potential of AD-hMSC, adipogenic and osteogenic differentiation were induced. Lipid vesicle formation was visible after 21 days of stimulation with dexamethasone and insulin via oil red staining (Figure 2k). On the other hand, calcium deposits were observed after 21 days of stimulation with BMP-4, dexamethasone, and ascorbic acid (Figure 2m). Nonstimulated AD-hMSC showed no positive staining after 21 days of culture in a maintenance medium (Figure 2j,l).

Advances in the therapeutic potential of mesenchymal stromal cells require preclinical results in order to design protocols for applications in regenerative medicine. AD-hMSCs express the characteristic mesenchymal immunophenotypes CD73, CD90, CD105, and CD166 and lack expression of hematopoietic markers such as CD34, CD45, and HLA-DR, as was determined. Our results were consistent with reported expression of mesenchymal markers (≥95%) and hematopoietic markers (≤2%), as determined by the International Society for Cell and Gene Therapy [24]. Stem cells have capabilities for self-renewal and multilineage differentiation and have a high potential to regenerate lost tissue [25]. The nuclear marker Ki67 is expressed in vertebrate cells and has been used to assess transcriptional activation and cell proliferation. Self-renewal was confirmed because more than 50% of AD-hMSC expressed Ki67.

This is relevant since it has been recently reported that this protein does not participate in tumor proliferation [26]. For multilineage differentiation, we confirmed that the adipogenic and osteogenic responses of AD-hMSC were consistent with previous reports for mesenchymal cells isolated from bone marrow aspirates [27] and from lipoaspirates (liposuction) [28]. Taken together, these cellular and phenotypical characteristics confirm the identity and isolation of an enriched mesenchymal stromal cell population.

### 3.3. Chondrogenic Differentiation Induces Aggregation and Expression of Surface Cartilage Markers in AD-hMSC Monolayer Culture

To determine the chondrogenic potential of AD-hMSC, cultures were exposed to differentiation media supplemented with TGF-β1, insulin, dexamethasone, and ascorbic acid. After 21 days of stimulation, bright-field microscopy showed that nonstimulated AD-hMSC formed a continuous monolayer, and ECM was not stained by alcian blue (Figure 3a,b). However, chondrogenic stimulation induced aggregation of AD-hMSC, forming cartilaginous nodules that were positively stained by alcian blue (Figure 3c,d). Staining was intense in areas where the ECM was more densely packed. It was evident that differentiated cells surrounding cartilaginous nodules also secreted proteoglycans to a lesser extent as they started to aggregate (Figure 3d). It was also of interest to analyze the changes in the expression of specific cartilage surface markers after chondrogenic stimulation. Compared to nonstimulated AD-hMSC (Figure 3e), chondrogenic differentiation induced changes in CD49a from 2.82% to 40.6%; CD49e from 51.4% to 92.2%; CD54 from 9.66 to 37.2%; and CD151 from 45.1% to 75.8% (Figure 3f) in hMSC-derived chondrocytes. Most mesenchymal markers remained unchanged after chondrogenic stimulation (CD73 from 95.4% to 93.8%; CD90 from 98.9% to 98.5%); as well as hematopoietic markers (CD45 from 0.20% to 0.18%; and HLA-DR from 0.16% to 0.31%).

Chondrogenic differentiated AD-hMSC maintained the expression of mesenchymal markers CD73 (93.8%) and CD90 (98.5%) after 21 days, compared to those observed in nonstimulated AD-hMSC, where expression was 95.4% and 98.9%, respectively. These markers have been associated with purine salvage pathways and membrane protein interactions and are highly expressed in cells surrounding capillaries and blood vessels [29]. As for surface markers in cartilage, analysis of nonstimulated AD-hMSC cultured in monolayer for 21 days revealed that mesenchymal cells expressed CD54 (9.66%), CD151 (45.1%), and CD49e (51.4%). This can be attributed to the fact that hMSC are considered chondroprogenitor cells [30], and for this reason, expression of these molecular markers is plausible. It is not clear whether CD49a can be considered a specific marker for active chondrocytes or for the chondrogenic differentiation of hMSC [31], although, expression in auricular chondrocytes is low (5%). However, our results revealed that CD49a expression changed from 2.82% in nonstimulated AD-hMSC to 40.6% in chondrogenically differentiated cells. It might be that this marker is expressed during chondrogenic differentiation, and its expression might be lost at more mature stages. For CD99 (transmembrane glycoprotein p30/32^mic2^) expression remained unchanged from 1.56 to 0.42% in chondrogenic cells. This is relevant since it has been reported that CD99 is highly expressed in malignant mesenchymal chondroblasts [32]. Minimal expression of CD99 in nonstimulated AD-hMSC was not altered by chondrogenic differentiation stimulation after 21 days. Expression levels remained less than 2% for both conditions (1.56% nonstimulated and 0.42% under chondrogenic conditions). Notably, expression of these cartilage markers increased after chondrogenic stimulation with TGF-β1, supporting the possibility of considering these molecules as biomarkers for hMSC-derived chondrocytes, as has been reported for chondroprogenitors and human articular chondrocytes [33].

### 3.4. Steam-Sterilized Gel/CS/PVA Hydrogel Is Biocompatible for the Culture of AD-hMSC

Once we determined the expression of cartilage markers after chondrogenic stimulation, we decided to evaluate the chondrogenic potential of AD-hMSC on steam-sterilized Gel/CS/PVA hydrogels. First, in order to determine the biocompatibility, nonstimulated cells were cultured on steam-sterilized hydrogels. AD-hMSCs were evenly distributed onto the hydrogel surface while the culture medium absorbed and hydrated the biomaterial. On day 2, individual cells adhered to the rugose surface and maintained fibroblastic morphology (Figure 4a). After 15 days, AD-hMSC formed a continuous monolayer, and cells established interconnections through extended filopodia. Cell morphology shifted from spiculated and cylindrical (day 2) to spiculated and flattened (day 15), with a visible increase in ECM deposition (Figure 4b,c). The viability assay showed that 87% of AD-hMSCs were metabolically active, as indicated by the calcein-AM fluorescent signal after 15 days of culture, while 13% were reported as dead (Figure 4d–g).

The combination of mesenchymal stromal cells and hydrogels is a strategy for developing treatments for cartilage regeneration. AD-hMSCs are a suitable cell type that has been proven to interact with diverse biocompatible scaffolds. The Gel/CS/PVA hydrogel provided biochemical components to stimulate the viability of hMSCs and eventual differentiation towards the chondrogenic lineage [34]. There are diverse factors that can affect the biological properties of cells. Initial tests evaluated the biocompatibility of Gel/CS/PVA hydrogel with the human colonic adenocarcinoma cell line HT29-MTX-E12 (ECACC 12040401); however, we determined that seeding onto Gel/CS/PVA hydrogels was found to be nontoxic and biocompatible with AD-hMSC (87% viability) as well as with hMSC-derived chondrocytes (99%).

### 3.5. Mesenchymal-Derived Chondrocytes Increase Chondrogenic Potential during Culture in Gel/CS/PVA Hydrogels

Once we confirmed the biocompatibility of AD-hMSC cultured onto Gel/CS/PVA hydrogels, we sought to determine whether this effect prevailed in culture conditions under chondrogenic stimulation. For this reason, nonstimulated AD-hMSCs were seeded onto steam-sterilized hydrogels and supplemented with standard media. Cells spread across the surface, adhered, and started to extend spiculated cytoplasmic projections (Figure 5a).

On day 1, standard media was replaced by chondrogenic differentiation media supplemented by TGF-β1, insulin, dexamethasone, and ascorbic acid. Over time, AD-hMSC started to migrate and condense into incipient nodules. On day 15, cells started to aggregate at several points throughout the monolayer and started to form chondrogenic nodules on the steam-sterilized GE/CS/PVA hydrogel, which increased in number as they became more cellularized in the periphery (Figure 5b). The formation of numerous chondrogenic nodules was evident. On day 21, chondrogenic nodules displayed increased size and density. At this point, a high number of the referred hMSC-derived chondrocytes migrated from the monolayer to become part of the chondrogenic nodules (Figure 5c). Cell viability remained at 99% and 1% mortality during chondrogenic stimulation from day 7 through day 21 (Figure 5d–f). The calcein assay allowed visualization of the aforementioned cell migration from the monolayer to chondrogenic nodules. Scanning electron microscopic analysis revealed that mesenchymal-derived chondrocytes adhered to the surface of the steam-sterilized Gel/CS/PVA with extended filopodia-secreted ECM fibers and displayed a slight increase in size and morphological changes. Polygonal morphology was evident on individual cells, especially in those located around chondrogenic nodules (Figure 5g), compared to the spiculated fibroblastic shape observed in nonstimulated AD-hMSC. Noteworthy, it was confirmed that chondrogenic nodule aggregation started on day 15 and was evident as the number of nodules anchored to the surface of the hydrogel increased. Surrounding hMSC-derived chondrocytes ensheathed the nodules or interacted with them through the formation of filopodia (Figure 5h,i). Immunochemistry analysis was performed to determine the constitution of chondrogenic modules after 21 days of induction. We determined that chondrogenic nodules were primarily composed of 29% aggrecan (Figure 5j,k) and 40.19% type II collagen (Figure 5l,m), two of the most abundant components in cartilage ECM.

Articular cartilage is a type of connective tissue with a balanced combination of water, proteins, and polysaccharides. Mesenchymal condensation is a crucial process for cartilage development, a process known as chondrogenesis, for which high cell density and three-dimensional conditions are vital. Condensation is one of the earliest events in chondrogenesis; it is promoted through cell-cell and cell-hydrogel interactions and starts with the aggregation of chondroprogenitor mesenchymal cells into precartilage aggregates [35]. We determined that interactions established between AD-hMSC and Gel/CS/PVA hydrogels under chondrogenic conditions accelerated in vitro mesenchymal condensation; however, further analysis should be performed to determine the expression of fibroblast growth factor (FGF, types 2, 4, 8, and 10), bone morphogenetic protein (BMP, types 2, 4, and 7), Sox9, fibronectin, and N-cadherin, among others.

Gel/CS/PVA hydrogel composition and adequate cell density contributed to an increase in mesenchymal condensation. As described, cell-to-cell interactions are both influenced by Gel/CS/PVA material composition and by physical contact as chondrogenic differentiation takes place. We determined that 1 × 10^4^ cells/cm^2^ for viability analysis and 4 × 10^4^ cells/cm^2^ for chondrogenic differentiation were optimal seeding densities for the Gel/CS/PVA hydrogel. This is relevant since other studies have reported that cell densities below the order of 12 × 10^6^ cells/mL were suboptimal for chondrogenesis and that this process was only favored by higher densities [36]. This has been previously demonstrated for alginate/gelatin hydrogels [37] and GAG-functionalized PEG hydrogels.

We also determined relevant changes in the expression of chondrocyte surface markers. For tetraspanin CD151, expression shifted from 45.1% in nonstimulated hMSC to 75.8% after 21 days of chondrogenic stimulation; intercellular cellular adhesion molecule 1 (ICAM-1, CD54) changed from 9.66% to 37.2%, and integrin α1β1 (CD49e) changed from 51.4% to 92.2%, respectively. Integrin α1β1 (CD49a) expression shifted from 2.82% in nonstimulated hMSC to 40.6% after chondrogenic differentiation. The increase in expression will be understood if we consider α1β1 as a chondrogenic marker for hMSC. This might also explain the morphological changes in hMSC-derived chondrocytes observed by SEM. The biological functions of these markers are associated with interactions with collagens found in ECM (integrin α1β1), recognition of RGD motifs with fibronectin (integrin α1β1), cell-ECM interactions (ICAM-1), cell integrity, and cell-cell interactions (tetraspanin); this is, proteins associated with chondrocytes related to other cartilage molecules found in ECM. Chondrogenic differentiation induced a remarkable increase in integrin expression. One example is integrin α1β1, which has increased expression in chondrogenic differentiation and is an important integrin related to cell migration and cell survival through its interaction with collagens. Additionally, integrin α5β1 is responsible for the recognition of RGD motifs found in fibronectin (cartilage ECM), which exerts an effect on cell migration, invasion, and proliferation. Another adhesion molecule is ICAM-1, which is expressed in chondrocytes and favors cell-cell and cell-ECM interactions. The membrane protein of the tetraspanin superfamily (CD151) also increased in expression in hMSC-derived chondrocytes and is involved in the maintenance of cell integrity, cell-to-cell communication, and cell motility.

Finally, the formation of chondrogenic nodules and the synthesis of cartilage ECM proteins promoted the expression of aggrecan and type II collagen. Aggrecan is a proteoglycan rich in chondroitin sulfate and keratin sulfate and provides mechanical support against compressive load. Type II collagen is a fibrillar protein that promotes the formation of ECM, constitutes 60% of the tissue dry weight, and functions as an anchorage point for fibronectin and collagens [38]. Our results suggest that interactions between hMSC-derived chondrocytes and Gel/CS/PVA hydrogels enhance chondrogenesis and speed up the expression of cartilage markers and ACAN and Col2; which are altogether mediated by the active participation of integrin signaling. These results match the expression profile expected for chondrocytes and support the multipotent capacity (trilineage differentiation) of adipose-derived mesenchymal stromal cells.

## 4. Conclusions

The chondroinductive properties of the Gel/CS/PVA hydrogel influence mesenchymal condensation, along with the use of soluble factors in chondrogenic stimulation media. The result is the acceleration of mesenchymal condensation of AD-hMSC, the subsequent increased differentiation of chondroprogenitor cells, and commitment to hMSC-derived chondrocytes. This research establishes the potential and plasticity of AD-hMSC for culture and differentiation on Gel/CS/PVA hydrogels. The properties of this material in combination with AD-MSCs make it a strong candidate for cartilage regeneration of and development of wound dressings.

## Figures and Tables

**Figure 1 polymers-15-03938-f001:**
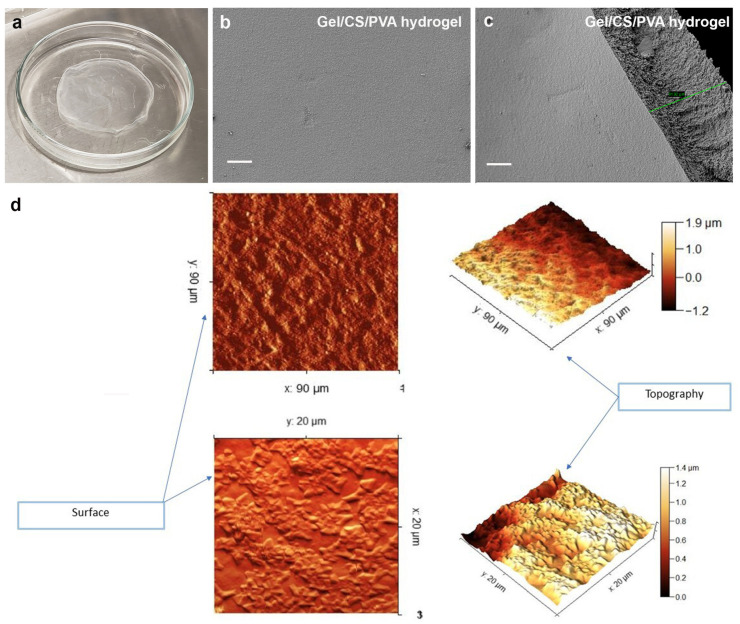
***Gel/CS/PVA hydrogel synthesis.*** A 2.5% *w*/*v* Gel solution was prepared. CS was dissolved in a 3% *v*/*v* acetic acid solution at a concentration of 2.5% *w*/*v*. PVA polymer was dissolved in distilled water until a 2.5% *w*/*v* solution was obtained. All the prepared polymer solutions were mixed in a 1:1:1 (*w*/*w*) polymer ratio of Gel:CS:PVA, respectively. Hydrogels were irradiated using ultraviolet radiation at 254 nm for 60 min. Finally, the hydrogels were sterilized in phosphate-buffered saline (PBS) using a steam autoclave SE 510. SEM images of Gel/CS/PVA hydrogel. The membranes showed thicknesses between 20 µm and 81 µm (**a**–**c**). Two-and three-dimensional AFM images of hydrogels top surface (**d**). Green line measure corresponds to 81.92 µm (**c**).

**Figure 2 polymers-15-03938-f002:**
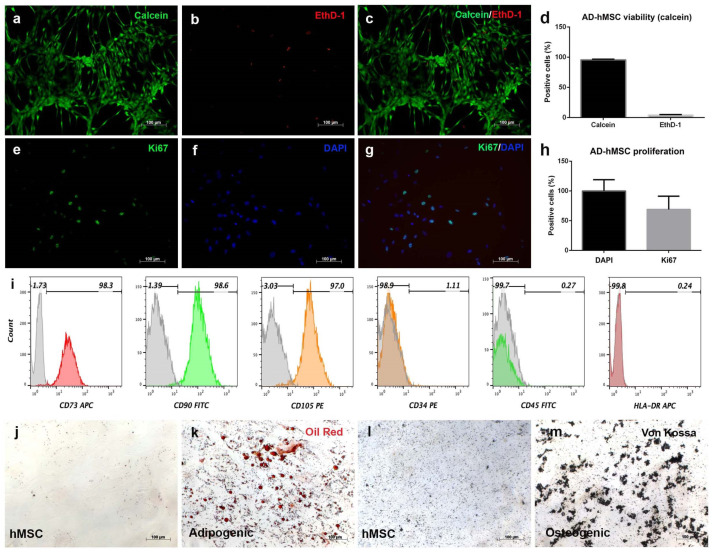
***Characterization of AD-hMSC.*** Mesenchymal stromal cells were isolated from adipose tissue and expanded in a monolayer for 15 days. Cells displayed 95.78% viability, 4.2% mortality (**a**–**d**), and 68.9% proliferative capacity (**e**–**h**). The mesenchymal expression profile was confirmed for the markers CD73, CD90, and CD105, while the expression of hematopoietic markers was not detected. Colored areas correspond to positive fluorescent signal for each marker, while gray areas correspond to cell autofluorescence (**i**). Adipogenic (**k**) and osteogenic (**m**) differentiation were confirmed in comparison with nonstimulated cells after 21 days (**j**,**l**). Scale bars 100 µm.

**Figure 3 polymers-15-03938-f003:**
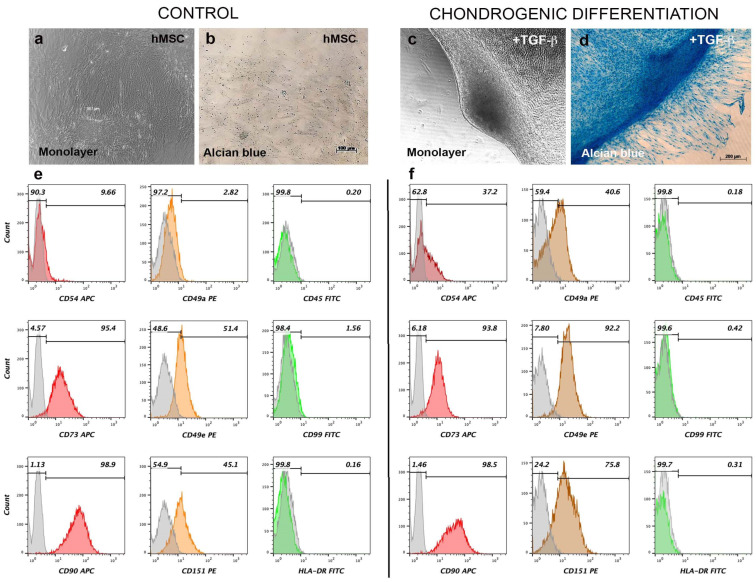
***Expression of cartilage surface markers in chondrogenically differentiated AD-hMSC.*** Confluent hMSC (**a**,**b**) or chondrogenic-differentiated cells (**c**,**d**) were harvested. A total of 2.5 × 10^5^ cells were labeled using 1.5 µg of CD marker antibodies and incubated for 30–45 min at 4 °C. The mesenchymal phenotype was detected using flow cytometry. Colored areas correspond to positive fluorescent signal for each marker, while gray areas correspond to cell autofluorescence (**e**,**f**, respectively). Alcian blue staining revealed the synthesis of a proteoglycan-enriched ECM typical of cartilage, thus confirming mesenchymal stemness.

**Figure 4 polymers-15-03938-f004:**
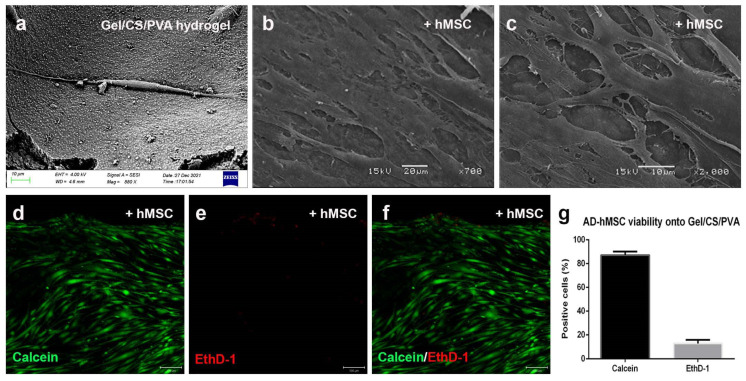
***Culture of AD-hMSC on Gel/CS/PVA hydrogels***. AD-hMSC were seeded onto 15-mm-diameter steam sterilized Gel/CS/PVA hydrogels at a density of 4 × 10^4^ cells/cm^2^. By day 1, AD-hMSC displayed fibroblastic morphology (**a**) and formed a continuous monolayer after 15 days culture onto Gel/CS/PVA hydrogels (**b**,**c**). Calcein-AM fluorescent assay showed that AD-hMSCs were metabolically active (**d**–**f**). Cell viability was 87.16% of the while 12.83% were reported to be dead (**g**). Scale bars correspond to 200 μm (**d**–**f**).

**Figure 5 polymers-15-03938-f005:**
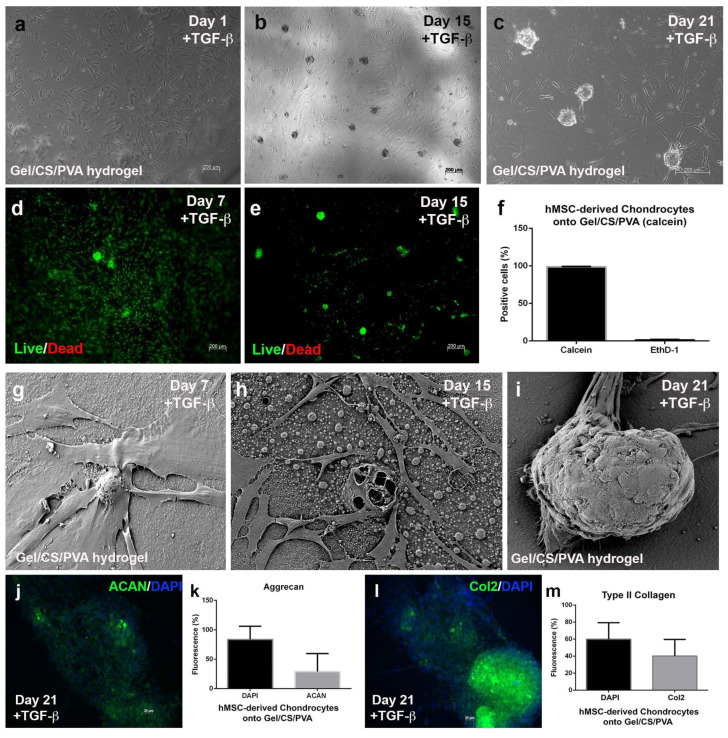
***Differentiation of AD-hMSC onto Gel/CS/PVA hydrogel***. AD-hMSC were seeded onto 15-mm-diameter steam-sterilized Gel/CS/PVA hydrogels at a density of 4 × 10^4^ cells/cm^2^. After 24 h, basic supplemented media was replaced with fresh chondrogenic induction media (DMEM-F12, 10 ng/mL TGF-β1, 1.0 mg/mL insulin, 10 mM dexamethasone, 20 mg/mL ascorbic acid, 1% FBS, and 1% penicillin-streptomycin). AD-hMSC displayed fibroblastic morphology by day 1 (**a**). Chondrogenic nodules started formation after 15 days stimulation (**b**) with number and size increase after 21 days (**b**,**c**). Calcein assay shows metabolically active cells in chondrogenic nodules (**d**,**e**) of 99% viability and 1% mortality (**f**). SEM micrographs revealed changes in cell morphology and formation of chondrogenic nodules (**g**–**i**). Expression of cartilage marker aggrecan (ACAN, 29%) and type II collagen (COL2, 40%) was analyzed in chondrogenic nodules (**j**–**m**).

## Data Availability

The data presented in this study are available on request from the corresponding author. The data are not publicly available due to intellectual property reasons.

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
