# Peer review of "Chondrogenic Potential of Human Adipose-Derived Mesenchymal Stromal Cells in Steam Sterilized Gelatin/Chitosan/Polyvinyl Alcohol Hydrogels"

_polymers, 2023, doi:10.3390/polym15193938_

Round 1

Reviewer 1 Report

Comments: 

1.       In the introduction, line 67 it was written "Gel-CS hydrogels technique" . Please remove the technique and re-write using the formulation, strategy, platform or even system

2.       In the introduction, Line 50, it was written “pharmaceuticals”. Please re-write to be pharmaceutical formulation

3.       Authors did not investigate TGF β and α SMA as main points for cell adhesion

4.       Cytoskeleton morphology was not also studied

Few English typos are captured in the text. Please revise them thoroughly 

Author Response

Reviewer 1 (Please see the attachment)

Comments: 

  1. In the introduction, line 67 it was written "Gel-CS hydrogels technique". Please remove the technique and re-write using the formulation, strategy, platform or even system.

Reply: Thank you for the thorough revision of the introduction. You are right. The sentence was corrected. The new sentence indicates that the polyelectrolyte Gel-CS hydrogels strategy was employed to formulate an optimized platform for 3D bioprinting and implement the resulting constructs in skin tissue engineering for wound healing. Changes were highlighted in red.

  1. In the introduction, Line 50, it was written “pharmaceuticals”. Please re-write to be pharmaceutical formulation.

Reply: Thank you for the thorough revision of the introduction. You are right. The sentence was corrected as you have requested. Changes were highlighted in red.

  1. Authors did not investigate TGF β and α SMA as main points for cell adhesion.

Reply: Thanks for the observation. We are currently working on a refined strategy to investigate the specific effect of substrate composition over the expression of α-smooth muscle actin (α-SMA) in focal adhesions established between AD-hMSCs or hMSC-derived chondrocytes and the Gel/CS/PVA hydrogel. Specifically, because it has been described that TGF-β stimulation and low serum concentrations increased expression of α-SMA in fibroblasts. As described, our chondrogenic differentiation protocol includes both, stimulation with 10 ng/mL TGF-β and low serum concentration (1% FBS) for 21 days. At this stage, we were determined to analyzed changes in expression of surface markers and expression of typical EMC proteins in cartilage. However, subsequent analysis will focus on detailed characterization for cell adhesion.

  1. Cytoskeleton morphology was not also studied.

Reply: Thanks for the observation. For the purposes of this study, the cytoskeleton was not the point of analysis. Although cells are directly influenced by direct contact with some polymeric network, such as the extracellular matrix or some synthetic scaffold, which can change some complex functions such as motility or differentiation, our experiments allow us to observe that Human Adipose-Derived Mesenchymal Stromal Cells and induced chondrogenic cells retain their phenotype and cell line potential when seeded on gelatin (Gel), chitosan (CS), and synthetic poly (vinyl alcohol) (PVA) hydrogels. We consider that the study of the cytoskeleton could be carried out in future experiments if we want to observe changes or effects in any signaling pathway that is of interest for the chondrogenic induction of stromal cells.

Comments on the Quality of English Language: Few English typos are captured in the text. Please revise them thoroughly.

Reply: Thank you for the thorough revision of the manuscript. You are right. The English typos where corrected as you have requested. Changes were highlighted in red.

Reviewer 2 Report

Lines 24-25, 48-49. “Copolymer hydrogels from natural compounds, namely gelatin (Gel), chitosan (CS), and  synthetic poly (vinyl alcohol) (PVA), have received increasing scrutiny because of their versatility, “The term "copolymer" refers to a polymer molecule composed of different monomeric moieties. Here we are talking about a mixture of three different polymers. Thus, a mixture of gelatin, chitosan and polyvinyl alcohol is not a copolymer. Perhaps, after UV radiation exposure, such a blend can be called a cross-linked polymer blend.

Line 26. “Previously, Gel/CS/PVA [1:1:1] terpolymer”. A terpolymer is a chemical compound resulting from a polymer that has a molecular structure built mostly or completely from a large number of similar units bonded together (such as a complex resin). A terpolymer is a result of the copolymerization of three different monomers, and is sometimes used to prevent corrosion. Thus, this mixture is not a terpolymer either! Lyophilization is a physical drying process that does not result in the formation of new chemical bonds, which is required to form a terpolymer from monomers.

Line 31. “95.78% viability”. What is the variation in this parameter from experiment to experiment? Specifying the parameter with accuracy not to whole percent, but to their hundredth parts requires huge research.

Lines 51-52. “These cross-linked networks are known to be chemical or permanent hydrogels with various physical shapes.” The term "cross-linked" refers to the formation of covalent bonds that are not formed when gelatin and chitosan are mixed together.

Lines 54-55. “The alginate-CS hydrogel integrated with Gel microspheres prepared by the emulsion cross-linking method presented suitable antibacterial properties [3]”. The antibacterial properties are due to the presence of tetracycline in the microspheres. The way it is written by the authors misleads the reader.

Lines 77-79. “It has been combined with CS and Gel to produce a terpolymer hydrogel Gel/CS/PVA by gamma radiation-induced polymerization for wound dressing use [15].” A terpolymer is the result of copolymerization of three different monomers, not radiation-induced cross-linking of three polymers.

Line 82. “succinate acid” succinic acid.

Lines 141,143, 144, 153. “(Agar Scientific Essex, UK)”, “(AGB7340, Agar Scientific Essex, UK)”, “(Zeiss Oberkochen, Germany)”, “(CSC11, Silicon-MDT, Russia)” Please specify city of production.

Line 149. “(Luzeren)” Please specify city and country of production.

Lines 291-292. “Hydrogels were irradiated using ultraviolet radiation at 254 nm for 60 min.” It is necessary to specify what is the radiation intensity, what is the spectral width of the radiation source (from how many to how many nm there is ultraviolet radiation in the beam), what is the exact thickness of each sample of irradiated material.

Lines 347-354 and Figure 2. And also further below the text. “Compared to non-stimulated AD-hMSC (Fig. 3e), chondrogenic differentiation to mesen chymal stromal cell-derived chondrocytes (hMSC-derived chondrocytes) induced changes in CD49a from 2.82% to 40.6%; CD49e from 51.4% to 92.2%; CD54 from 9.66 to 350 37.2% and CD151 from 45.1% to 75.8% (Fig. 3f). Most mesenchymal markers remained unchanged after chondrogenic stimulation (CD73 from 95.4% to 93.8%; CD90 from 98.9% 352 to 98. 5%); as well as, haematopoietic markers (CD45 from 0.20% to 0.18%, and HLA-DR from 0.16% to 0.31%).” How reliable is it to bring values to the nearest hundredth of a percent? Are these really statistically significant values?

Lines 556-557, 562-567, 580-585. Text needs to be deleted or corrected.

Author Response

Reviewer 2 (Please see the attachment)

Comments and Suggestions for Authors

  1. Lines 24-25, 48-49. “Copolymer hydrogels from natural compounds, namely gelatin (Gel), chitosan (CS), and synthetic poly (vinyl alcohol) (PVA), have received increasing scrutiny because of their versatility, “The term "copolymer" refers to a polymer molecule composed of different monomeric moieties. Here we are talking about a mixture of three different polymers. Thus, a mixture of gelatin, chitosan and polyvinyl alcohol is not a copolymer. Perhaps, after UV radiation exposure, such a blend can be called a cross-linked polymer blend.

Reply: Thank you for the thorough revision of the abstract and introduction. You are right. The sentence was corrected as you have requested. Changes were highlighted in red.

  1. Line 26. “Previously, Gel/CS/PVA [1:1:1] terpolymer”. A terpolymer is a chemical compound resulting from a polymer that has a molecular structure built mostly or completely from a large number of similar units bonded together (such as a complex resin). A terpolymer is a result of the copolymerization of three different monomers, and is sometimes used to prevent corrosion. Thus, this mixture is not a terpolymer either! Lyophilization is a physical drying process that does not result in the formation of new chemical bonds, which is required to form a terpolymer from monomers.

Reply: Thank you for the detailed review of the abstract. I've made the corrections as per your request, with changes highlighted in red.

  1. Line 31. “95.78% viability”. What is the variation in this parameter from experiment to experiment? Specifying the parameter with accuracy not to whole percent, but to their hundredth parts requires huge research.

Reply: Viability assays were performed as follow: AD-hMSC were expanded and directly seeded onto 15 mm diameter steam-sterilized Gel/CS/PVA hydrogels at a density of 4 x 104 cells/cm2. Cell viability analysis consisted of individual quantification of live/dead cells from fluorescence micrographs. This result was expressed as percentage of viability or mortality. For AD-hMSC cultured in monolayer, cell viability was 95.78 % ± 0.88 (line 31). For AD-hMSC seeded onto Gel/CS/PVA hydrogels, viability was 87.16% ± 3.05% and for hMSC-derived chondrocytes was 98.7% ± 0.65 (line 421-423). According to these percentages, we found Gel/CS/PVA hydrogels to be nontoxic and biocompatible to AD-hMSC. Variation was expressed, as SD we have determined that there were differences between experiments.

  1. Lines 51-52. “These cross-linked networks are known to be chemical or permanent hydrogels with various physical shapes.” The term "cross-linked" refers to the formation of covalent bonds that are not formed when gelatin and chitosan are mixed together.

Reply: Appreciation for your comprehensive review of the introduction. You are right. As requested, I have made the necessary corrections, highlighting them in red.

  1. Lines 54-55. “The alginate-CS hydrogel integrated with Gel microspheres prepared by the emulsion cross-linking method presented suitable antibacterial properties [3]”. The antibacterial properties are due to the presence of tetracycline in the microspheres. The way it is written by the authors misleads the reader.

Reply: Thank you for the thorough revision of the introduction. You are right. The sentence underwent revisions to enhance clarity, and modifications were marked in red.

  1. Lines 77-79. “It has been combined with CS and Gel to produce a terpolymer hydrogel Gel/CS/PVA by gamma radiation-induced polymerization for wound dressing use [15].” A terpolymer is the result of copolymerization of three different monomers, not radiation-induced cross-linking of three polymers.

Reply: Thank you for the thorough revision of the introduction. You are right. The sentence was corrected as you have requested. Changes were highlighted in red.

  1. Line 82. “succinate acid” succinic acid.

Reply: Thank you for the thorough revision of the introduction. You are right. The word was corrected, and modification was marked in red.

  1. Lines 141,143, 144, 153. “(Agar Scientific Essex, UK)”, “(AGB7340, Agar Scientific Essex, UK)”, “(Zeiss Oberkochen, Germany)”, “(CSC11, Silicon-MDT, Russia)” Please specify city of production.

Reply: Thanks for the comment. The information was added in the manuscript.

Lines 141, 143 and 144 “… electrically conductive carbon double-sided adhesive discs (Agar Scientific. Stansted, Essex, UK). , The samples were coated with gold with two thin layers in two cycles of 20 sec each by plasma-assisted deposition using a manual sputter coater (AGB7340, Agar Scientific Stansted, Essex, UK). The images were acquired with a FIB-SEM Crossbeam 550 field emission microscope Zeiss (Oberkochen, Baden-Württemberg, Germany) at 4 kV.”

Line 153

“… with silicon tips (CSC11, Silicon-MDT, Moscow, Russia) taking images at a scanning rate of 3 lines/s…”

  1. Line 149. “(Luzeren)” Please specify city and country of production.

Reply: Thanks for the comment. The information was added in the manuscript.

Gel/CS/PVA hydrogel films were dehydrated at a controlled temperature of 50 °C in a convection oven (Luzeren, Hangzhou, Zhejiang. China)

  1. Lines 291-292. “Hydrogels were irradiated using ultraviolet radiation at 254 nm for 60 min.” It is necessary to specify what is the radiation intensity, what is the spectral width of the radiation source (from how many to how many nm there is ultraviolet radiation in the beam), what is the exact thickness of each sample of irradiated material.

Reply: Thanks for the comment. The information was added in the manuscript.

“Hydrogels were irradiated using ultraviolet radiation at 254 nm at 120,000 µJ/cm2 for 60 min in a CL-1000 UV Crosslinker (Cambridge, England).

  1. Lines 347-354 and Figure 2. And also further below the text. “Compared to non-stimulated AD-hMSC (Fig. 3e), chondrogenic differentiation to mesenchymal stromal cell-derived chondrocytes (hMSC-derived chondrocytes) induced changes in CD49a from 2.82% to 40.6%; CD49e from 51.4% to 92.2%; CD54 from 9.66 to 350 37.2% and CD151 from 45.1% to 75.8% (Fig. 3f). Most mesenchymal markers remained unchanged after chondrogenic stimulation (CD73 from 95.4% to 93.8%; CD90 from 98.9% 352 to 98. 5%); as well as, haematopoietic markers (CD45 from 0.20% to 0.18%, and HLA-DR from 0.16% to 0.31%).” How reliable is it to bring values to the nearest hundredth of a percent? Are these really statistically significant values?

Reply: We employed flow cytometry for evaluating percentage of expression of mesenchymal and cartilage cell surface markers. Both software’s, Cell Quest software for acquisition of data and Flow Jo for analysis of data, have the same predefined precision and accuracy. Results from this analysis are expressed with two digits after the decimal point. For this reason, data are presented with this precision. The precision for reporting of each statistic depends on how that statistic is derived(1). Values presented at this level of accuracy should not affect statistical differences. It is important to consider that these values can also be expressed as absolute cell count, according to the acquired number of events (10,000 events). We are reporting the percentage of those events that were positive for the specific markers.

  1. Habibzadeh F, Habibzadeh P. How much precision in reporting statistics is enough? Croat Med J. 2015;56(5):490–2.
  2. Lines 556-557, 562-567, 580-585. Text needs to be deleted or corrected.

Reply: Thank you for the thorough revision. You are right. The text was deleted and additional info as you have requested. Changes were highlighted in red.
